# Feasibility Assessment of the Let’s Walk Programme (CAMINEM): Exercise Training and Health Promotion in Primary Health-Care Settings

**DOI:** 10.3390/ijerph18063192

**Published:** 2021-03-19

**Authors:** Sebastià Mas-Alòs, Antoni Planas-Anzano, Xavier Peirau-Terés, Jordi Real-Gatius, Gisela Galindo-Ortego

**Affiliations:** 1National Institute of Physical Education of Catalonia (INEFC), Lleida Campus, E-25192 Catalonia, Spain; aplanas@gencat.cat (A.P.-A.); xpeirau@gencat.cat (X.P.-T.); 2Grup de Recerca de Moviment Humà, University of Lleida (UdL), E-25192 Catalonia, Spain; 3School of Medicine and Health Sciences, Universitat Internacional de Catalunya, E-08195 Sant Cugat del Vallés, Spain; jreal@idiapjgol.info; 4Unitat de Suport a la Recerca Lleida—Barcelona, Institut Universitari d’Investigació en Atenció Primària Jordi Gol (IDIAP Jordi Gol), Lleida, E-25007 Catalonia, Spain; 5Catalan Institute of Health, Lleida, E-25002 Catalonia, Spain; ggalindo.lleida.ics@gencat.cat

**Keywords:** physical activity, RE-AIM, HEPA, community-based participatory research, adherence, health outcomes, exercise prescription, implementation science, public health, sedentary behavior

## Abstract

Exercise is related to many individual health outcomes but impact evaluations of exercise programmes are seldom conducted. The purpose of the study is to evaluate the feasibility of an exercise prescription intervention in primary health-care settings (CAMINEM Programme) located in two socially disadvantaged neighbourhoods. The CAMINEM was a pragmatic-driven intervention with opportunistic recruitment. It followed the 5As framework for health promotion and also the exercise training principles. Feasibility was evaluated using the RE-AIM framework (Reach, Effectiveness, Adoption, Implementation, and Maintenance). Patients with non-communicable chronic diseases participated in a 12-month home-based moderate-intensity exercise program, counselled by exercise physiologists. Participants were grouped according to their physical activity behaviour at baseline and 6-month adherence. CAMINEM reached 1.49% (*n* = 229) of the eligible population (*N* = 15374) and included a final sample of 178. Health outcomes for adhered participants followed positive patterns. Non-adhered participants visited their practitioner more compared to adhered participants. Thirty-three practitioners (40%) referred patients. Nurses referred four times more than physicians (81% and 19% respectively). The delivery of exercise prescriptions proved to be easy to complete and record by participants as well as easy to monitor and adjust by the exercise physiologists. One out of four participants adhered during the 12-month intervention. This intervention has been feasible in primary care in Catalonia, Spain, to safely prescribe home-based exercise for several conditions.

## 1. Introduction

Physical inactivity was identified as the fourth leading risk factor for global mortality [1] and trend data showed limited improvement between 2001 and 2016 [2] and between 2007 and 2017 [3]. Consequently, member states of the WHO agreed to achieve a 10% relative reduction in the prevalence of physical inactivity by 2050, as one of the global targets for the prevention and treatment of non-communicable diseases [4]. They also suggested that some differences in physical activity levels within and between countries can be explained by inequities in opportunities to be physically active, as was identified by Bauman et al. [5]. Even low levels of physical activity may reduce morbidity, all-cause mortality, and length life expectancy [6]. Accordingly, physical activity promotion is already issued in clinical guidelines, even excluding physical activity-specific guidelines [7]. Health-enhancing physical activity (HEPA) promotion includes primary health-care (PHC) settings because they reach a substantial number of people who overall, are more inactive and may benefit more [8,9]. In Spain, roughly 46% of the population are advised to walk as HEPA promotion [10].

Practitioners need systematic work to achieve habit change such as sedentary behaviour [11]. Written physical activity prescriptions have been feasible for practitioners and patients in health-care settings [8,12,13]. However, Spanish interventions of HEPA are seldom detailed and reported [10]. Catalan general practitioners rarely see HEPA promotion as a priority in 5-min consultations [14]. Moreover, 55% of the physicians who already promote HEPA consider there to be not enough time. Recently, Martínez–Gómez et al. [10] suggested that Spanish physicians may tailor counselling and provide written directions of HEPA, but there is a challenge to structuring this counselling (e.g., frequency and duration of walking) given time constraints and lack of staff training. Additionally, pre-exercise medical clearance is generally unnecessary [15], especially when the intensity of the activity is planned to be light to moderate [16].

Feasibility of the approaches is not universal; policies and legal frameworks differ between countries, and even between regions within the same country (e.g., Spain). The Reach, Efficacy/Effectiveness, Adoption, Implementation, and Maintenance (RE-AIM) framework [17] was first designed to estimate the public health impact of an intervention and it has been accepted to target the feasibility of HEPA interventions in community settings, such as PHC centres [18,19].

Recently, the European Commission, through the Public Health Agency, made a call for countries to transfer the Swedish Physical Activity on Prescription Model to other European regions, and the EUPAP Project (www.eupap.org, accessed on 1 February 2021) is actually ongoing and aiming to implement the Swedish Physical Activity on Prescription Model into nine other countries: Catalonia (Spain), Denmark, Flanders (Belgium), Germany, Italy, Lithuania, Malta, Portugal, and Romania [20]. The health sector is the gate for citizens to receive HEPA prescriptions, mostly primary healthcare settings, and the first option is that the physical activity is performed outside the healthcare setting. The core components of the Swedish model to be adapted by the EUPAP Project include: individualised patient-centered counselling, written prescription, evidence-based physical activity recommendations, follow-up, and collaboration with activity organisers [21].

The overall aim of this pragmatic study was to evaluate the feasibility (i.e., external validity) of the CAMINEM (Let’s Walk) Programme as an exercise-on-prescription, interdisciplinary approach for 12 months in two socially disadvantaged neighbourhoods. The specific aims were to (i) determine if CAMINEM reached the citizens with chronic health conditions that may benefit from exercising (Reach), (ii) assess clinical and quality-of-life outcomes and the use of health services (Effects), (iii) determine the adoption of CAMINEM by the practitioners (Adoption), (iv) evaluate participants’ retention and compliance (Implementation), and (v) describe if the practitioners prescribe exercise post-intervention (Maintenance).

## 2. Materials and Methods

The study was a pragmatic-driven intervention with opportunistic recruitment by general practitioners (GPs) or community nurses during 18 months–that is, regular patients aged 18 and above who visited their GP or community nurse may be invited. Participants received an exercise written prescription and were followed-up by an exercise physiologist for twelve months before being discharged. The rationale and design of the CAMINEM programme, including detailed procedures (referral, first face-to-face contact, follow-up, discharge) is published elsewhere [22,23]. Appendix A illustrates them.

There were no fixed inclusion criteria for the patients but practitioners focused on diseases in which exercise is known to be beneficial (i.e., overweight/obesity, hypertension, diabetes mellitus, dyslipidaemia, musculoskeletal pain, respiratory diseases, and minor mental health problems). Exclusion criteria were overt cardiovascular disease, uncontrolled hypertension, uncontrolled insulin-dependent diabetes, or other conditions that prevented participation in a walking programme determined by the practitioner or the exercise physiologist.

The settings of intervention were selected following a convenience sampling; that is, two primary health-care settings in the city of Lleida that offered health services to the two highest socially disadvantaged neighbourhoods, out of a total of seven primary health-care settings. The reason for this is that Catalan health policies encourage health promotion in such neighbourhoods, and it has eventually been found positive by WHO [24,25]. All GPs and community nurses working at least 3 months in any of these two settings were invited to participate by referring patients to the CAMINEM Programme.

The CAMINEM Programme provides a set of tools for practitioners to individualise exercise prescriptions: validated urban routes to monitor HEPA [26], a short questionnaire to asses physical activity level (ClassAF) [27], and a prescription form/logbook to monitor adherence (i.e., exercising following the prescriptions). This pragmatic study included the participation of a non-staffed exercise physiologist who counselled the participants previously referred by their GP or nurse. Exercise counselling relied on the 5As framework (Ask, Advise, Agree, Assist, Arrange) for health promotion [28] and the exercise training principles. Exercise periodization followed the health-oriented guidelines from the American College of Sports Medicine [29] and the Catalan exercise prescription handbook [27]. The conditioning period goal (up to two months) was to ensure that participants followed the WHO recommendations on physical activity for health for adults and older adults with a minimum of 150 minutes weekly of moderate-intensity aerobic physical activity [1]. The improvement period (two to six months) focused on increasing the total amount of exercise volume (first frequency, then duration), and on keeping adherence. The maintenance period (six to 12 months) aimed to maintain or increase exercise volume.

The exercise was unsupervised, individually based, moderate-intensity, continuous, and aerobic; that is, walking. Individualization of prescriptions was set during the first meeting with the exercise physiologist who underwent a motivational interview with each participant. Together they set goals, chose the specific urban route, set the frequency and duration of each walk, and scheduled the next follow-up meeting. The exercise physiologist delivered the written exercise prescription and then explained how to self-control the intensity of exercising using the Talk Test [30], as it has recently been suggested [31]. Follow-up meetings were for monitoring the compliance (adherence), giving advice to overcome possible barriers, and discussing the exercise prescription to finally deliver a new update on the written exercise prescription. Participants completed the full intervention after 12 months, were discharged and encouraged to keep an active lifestyle, and monitor their HEPA. Specific criteria for decision making are shown in Supplementary Material S5.

For the feasibility study we evaluated the five dimensions of the RE-AIM framework [17](see Table 1).

Reach referred the participation rate among eligible patients, taken from the health provider medical records. Reasons for exclusion and dropout were collected in a Microsoft^®^ Access database created ad hoc.

Effects outcomes included clinical health, health-related quality of life (HRQOL), and health services use. Clinical health outcomes (BMI, waist circumference, blood pressure, resting heart rate, and biochemical compounds from blood samples [glucose, triglycerides, glycated haemoglobin, HDL, LDL, and total cholesterol]) were collected before (M1 pre-test), during (M2 6 months), and after (M3 post-test) the intervention by nurses during their regular routine unless no data had been recorded within 6 months prior to the referral. To measure HRQOL we used the Short Form 12 Health Survey version 2 (SF-12v2) in Catalan [34], and a simple question comparing self-perceived well-being in two separate moments: ‘What do you think about your overall health, is it better, worse, or the same as the day you started the CAMINEM?’. Health services use (i.e., the number of monthly visits with practitioners) data were obtained from the health provider database, and was measured as a ratio of number of the visits to the physician or nurse per month. We focused on the effects and not efficacy/effectiveness because efficacy trials can be obtained under optimal conditions (e.g., RCT design) (Flay, 1986, as cited in Estabrooks & Gyurcsik, 2003, p. 43) and effectiveness is ”the extent to which the intended effect or benefits that could be achieved under optimal conditions are achieved in practice” [35]. Since none of the previous had been done before, we relied on the health effects.

Adoption referred as the participation rate among the total number of potential participant agents (i.e., staffed GPs and nurses).

Implementation referred to participants’ adherence, which included (i) retention days, (ii) attendance at follow-up meetings, and (iii) exercise compliance. For participants to be compliant they had to complete more than 50% of the prescribed exercise sessions. The 6-month assessment was considered a critical breakpoint since most dropouts occur within that time when starting a physical activity program [36]. Participants included longer than six months were considered as retained but not necessarily adhered to the intervention. They may attend follow-up meetings but may not exercise as intended (e.g., less frequently). Safety was measured as the number of adverse events reported by either participants or practitioners while exercising.

Maintenance data at the individual level were to be assessed 3-months post-intervention by regards of clinical, HRQOL and physical activity level. Assesments at 6- and 12-months post-intervention were set as the number of prescriptions delivered by the practitioners. Unfortunately, these assessments were not done due to a high amount of missing data that were supposed to be collected by regular practitioners.

Variables were analysed between 4 cohort groups distributed according to their basal physical activity level (AC/IN, previously active/inactive) and their adherence for a minimum of 6 months (AD/NA, adhered/non adhered). Dropout participants were evaluated as intention-to-treat. Categorical variables are presented as percentages. Pearson’s chi-squared (χ^2^) tests were used to determine differences between groups. Continuous variables are presented as the mean value and standard deviation (SD). Statistical differences were assessed by analyses of variance (ANOVA) by time in relation to the four groups using the Schefée test, a contrast coefficient test when comparing two subgroups out of all four, and the t-test when comparing two groups only (i.e., effects for the adhered groups only). Normally distributed variables were tested according the Shapiro–Wilk test for samples lower than 30, and the Kolmogorov–Smirnov test if higher than 30. Analyses pre- and post-intervention were done if the sample was higher than five. Continuous variables that statistically differed between groups at baseline (i.e., confounding factors) were used as covariates. ANOVA was used for variables following normal distribution. Non-parametric tests were applied for variables that were not normally distributed: the Mann–Whitney *U* test between groups, the Wilcoxon signed-rank test (Wilcoxon’s *Z*) for two measurements within groups, and the Kendall’s coefficient of concordance (Kendall’s *W*) for three measurements within groups. Mean differences between moments were shown as total difference and the percentage of difference. PASW statistics (release 18.0.0) was used for data treatment and for all analyses. Statistical significance was set at *p* < 0.05 with 95% of confidence intervals.

The Clinical Investigation Ethics Committee of the IDIAP, Jordi Gol, Barcelona, approved this study, and it followed the ethical principles of the Declaration of Helsinki. Participants gave their verbal consent for referral and signed an informed consent before inclusion. Clinical data collection was part of the ordinary healthcare practice and written permission to use these data was obtained from the healthcare provider. The exercise physiologists were members of the national exercise professional association in accordance with Catalan regulations.

## 3. Results

### 3.1. Reach

Figure 1 shows the flowchart of the 178 participants included in the study (mean age 58.1 ± 12.2, women *n* = 115), the number of retained participants (i.e., not dropout even without being compliant with the exercise prescriptions), participants who adhered to the program (i.e., were compliant with the prescriptions), and the reasons for dropout. Note that 19 participants were still adhering to the intervention by the time this feasibility study ended; that is, they did not complete the minimum of 6-month adherence but did not drop out either.

The most common reasons for referral were dyslipidaemia and diabetes mellitus in men, and musculoskeletal diseases and mental health conditions in women. Patients with high blood pressure were referred almost exclusively by nurses: 48% of patients suffering from it were referred by nurses, compared to 3% by GPs. Practitioners referred from one healthcare setting (PHC_A) mostly referred patients with one (45.5%) or two (31.2%) conditions, while practitioners from the other healthcare setting (PHC_B) referred patients with more than one condition (82.2%). The youngest age group (18 to 44 years) was mostly referred due to overweight (84%) and was diagnosed with one or two conditions only (88%). Patients from 45 to 64 years were referred due to respiratory diseases, mental ill-health, and other reasons in a greater proportion than other age groups. Almost one-third (31.1%) of older adults (65 years and above) were diagnosed with a cardiovascular condition, while the percentage in other age groups was below 10%. Participants reporting lower levels of PA were generally referred with more than one health condition, and the sufficiently active were referred with a lower number of conditions (see Table 2).

### 3.2. Effects

Complete collection of clinical parameters was obtained from 43 participants (24%), and 29 (16%) including HRQOL. Table 3 shows descriptive statistics for baseline clinical parameters, HRQOL and healthcare attendance among groups. Participant characteristics did not differ significantly when comparing two subgroups in relation to their basal PA behaviour. Although overweight participants were the majority in most groups, the AD-AC group (adhered and previously active) statistically showed a lower proportion (χ^2^ = 16.3, *p* = 0.001) of overweight participants than other groups. The most common profile for participants in all groups was a woman aged 45–64 being referred by her nurse.

Due to a low sample for four measurements, basal (M1) and post-intervention intention-to-treat follow-up (MOT3) were chosen for pre-post analyses and grouping adhered participants and non-adhered participants. Thus, the adhered group (AD-Group) comprised of AD-IN and AD-AC and the non-adhered group (NA-Group) comprised of NA-AC and NA-IN. Homogeneity between these two groups was calculated by contrast coefficient tests not assuming equal variances for age (*t* = 1.552, *p* = 0.126) and assuming them for BMI (*t* = 0.731, *p* = 0.466). As a result, adhered and non-adhered groups were homogeneous at baseline for all clinical parameters, as well as for categorical variables except those diagnosed with dyslipidaemia and overweight (see Table 4).

There were no statistical differences between the two groups on baseline data, neither for continuous variables nor for qualitative data (gender, practitioner referring, PHC centre) except those diagnosed with dyslipidaemia or overweight.

Comparative analyses within and between groups were done using ANOVA for variables following normal distribution, which resulted to be: resting heart rate, total blood cholesterol, and LDL-cholesterol. Non-parametric tests were applied for variables that were not normally distributed: body mass index, waist circumference, systolic and diastolic blood pressure, triglycerides, HDL-cholesterol, glucose, and glycated haemoglobin.

As shown in Table 5, most clinical parameters improved over time for both groups: participants who adhered and who did not adhere. Improvements in total blood cholesterol (*p* = 0.008, 85% CI [4.3, 26.9]) and BMI (*p* = 0.013) were found over time regardless of grouping. Adhered participants saw a significant decrease in their triglycerides level (28.77%, *p* = 0.009) when comparing means range. The NA-Group showed significant improvements in total blood cholesterol (11.71%, *p* = 0.001, 95% CI [9.7, 36.9]). The range mean for BMI also improved significantly for the NA-Group (1.91%, *p* = 0.002); although it did not for the AD-Group, most likely due to a larger sample. Differences in range mean for systolic blood pressure were also significant for the NA-Group (3.50%, *p* = 0.019) while it did not improve for the AD-Group.

After the intervention all participants answered feeling equal or better than the inclusion day. None of the respondents reported feeling worse compared to their first day and 100% of previously inactive participants reported feeling better than their first day. No participants (*n* = 0) reported any health problem due to the exercise prescriptions.

Participant health-care attendance was similar before, during, and after the CAMINEM intervention. Globally, participants averaged 1.5 visits per month in the year before their inclusion. Participants who finally adhered averaged 1.3 visits to their health practitioner during the time the CAMINEM was delivered, less than participants who did not adhere (1.5), although these differences were not statistically significant (*p* > 0.05). See Figure 2.

### 3.3. Adoption

A total of 82 health practitioners were invited to refer patients and none of them formally refused to participate, although 49 (60%) did not refer any patients throughout the 18-month intervention. Three patients registered at other primary health-care settings prompted their GP to be referred, and were finally included. Nurses referred 186 (81%) patients and GPs referred 43 (19%). Ten practitioners referred more than ten patients while another ten referred only one. No supply physicians referred patients, whereas nurses referred patients regardless of their employment situation: eight out of twelve substitute nurses referred (66.67%) and 19 out of 34 regular nurses (55.88%) did.

### 3.4. Implementation

Thirty-five participants (19.66%) adhered for more than 180 days. Twelve other participants were retained for more than six months without being compliant. Almost half of non-adhered participants (42.3%) followed the exercise prescription before dropout. Table 6 shows the distribution of adherence of the four groups. Adhered participants were of similar age compared to the non-adhered, although the AD-AC group was the oldest, with a mean value over 64 years and 95% CI [61.38, 67.52]. Ninety-eight participants (56.60%) dropped out within the first three months. The main reasons for dropping out were not attending three consecutive meetings or not answering three phone calls.

## 4. Discussion

This study was, to our knowledge, the first pragmatic study assessing the feasibility of an exercise prescription intervention in primary health-care settings based on matching exercise training principles and public health promotion. The feasibility of pragmatic interventions intends to answer the recommended evaluative questions suggested by Estabrooks & Gyurcsik across the RE-AIM dimensions [37]. The “efficacy/effectiveness” dimension was substituted by the “effects” due to lack of a previously efficacy trial (i.e., under optimal conditions).

The overall aim of this pragmatic study was to evaluate the feasibility (i.e., external validity) of the CAMINEM (Let’s Walk) Programme as an exercise-on-prescription, interdisciplinary approach for 12 months in two socially disadvantaged neighbourhoods. The specific aims were to (i) determine if the CAMINEM reached the citizens with chronic health conditions that may benefit from exercising (Reach), (ii) assess clinical and quality-of-life outcomes and the use of health services (Effects), (iii) determine the adoption of CAMINEM by the practitioners (Adoption), (iv) evaluate participants’ retention and compliance (Implementation), and (v) describe if the practitioners prescribe exercise post-intervention (Maintenance). As stated previously, maintenance could not be finally evaluated due to a low sample for data collection.

### 4.1. Did the CAMINEM Programme Reach the Target Population?

The results flagged up 16,744 patients who fulfilled the inclusion criteria, of which 15,374 (91.82%) had visited their health practitioner at least once during the intervention. The proportion is slightly higher than the average of 87% of Spanish citizens who visited their practitioners in 2005 [38], and similar to 94% of citizens who lived in the same region where the study was implemented, based on self-reported data from 2018-19 [39]. Selected nurses volunteered to record the number of patients who were invited, and who did not wish to take part in the study. However, they did not report any data. They were reminded several times but eventually the research group decided not to insist due to practitioners’ time constraints. If we could analyse the number of withdrawals at this stage, then we may drive the intervention to: more stress on behaviour counselling (if high number of patients refusing) or more emphasis on recruitment (if low number of invitations). The lack of a concrete number of target participants refusing to take part in the physical activity referral intervention was also reported in the EXERT study [40]. In contrast to the EXERT, all referrals (*n* = 229) reached the exercise physiologist, which is the 1.49% of the eligible population (*N* = 15374). Of referrals, 28 (12.23%) patients did not attend the first counselling meeting and the final number of included participants was 178, 1.16% of the eligible population.

A similar proportion is reported in other studies. Of all participants referred by a health professional, 27% never made contact with the exercise referral scheme in a British study [41]. More recently, data of seven years of the Welsh referral programme showed a reach of 3.3% of the population [42], which is an average of roughly 0.7% each 18 months, the length of the CAMINEM study. In Sweden, the FaR^®^ scheme reached 1.5% of the total population, and 1.3% of those who attended their primary health-care setting [43]. Less than 1% was reported in a revision in the UK [44]. More intensive interventions reached a higher proportion of the target population. One intervention in Switzerland aimed at increasing physical activity in inactive regular patients invited all patients attending five volunteer GP for an RCT [45]. The practitioners were reimbursed with 25 CHF (€ 18) for each questionnaire that was filled out in their office. The Newcastle exercise project reached 17% for their RCT, but the researcher was in the practice to initiate recruitment daily [46].

In sum, 1.16% of the eligible population reached the programme which is a similar proportion to other physical activity prescription intervention studies.

### 4.2. What were the Effects of the CAMINEM PROGRAMME?

Clinical health and quality-of-life outcomes for adhered participants followed positive patterns, although few were statistically significant. From a pragmatic point of view, note that non-adhered participants visited their practitioner more compared to adhered participants. Also, none of the participants reported any injury or complication of their disease due to the exercise participation, which let us confirm that the intervention was safe even without supervision by exercise professionals. This programme followed global and regional recommendations on health-enhancing physical activity [1,15,24,27] and its results could be useful for future interventions. A systematic review of exercise referral schemes did not find consistent evidence in favour of the interventions in outcomes based on health-related quality of life, blood pressure, serum lipid levels, indices of obesity, or glycemic control, among others [47].

Despite the aforementioned, every increase in physical activity, even small, entails health gains at both an individual level and public health level regardless of whether they reach the recommendations of physical activity levels [6,48].

### 4.3. Was the CAMINEM Programme Adopted by the Health Practitioners?

None of the practitioners formally refused to participate and nurses referred four times more than physicians (81% and 19% respectively). One GP and one community nurse form the primary health-care unit patients can visit. Nurses were more engaged in the intervention. However, nurses were in charge of collecting the clinical data at baseline, during and after the intervention but they did not collect as it was agreed due to time constraints. Regular referral by the 18-month timeframe of the implementation suggests that the CAMINEM procedures facilitated the Catalan practitioners to cope with the barriers for HEPA promotion [14].

### 4.4. Could the CAMINEM Programme Be Implemented the Way it was Designed?

Our challenge was to monitor exercise prescription adherence and collect reported data from the participants to adapt new exercise prescriptions following individualised progressions to achieve greater benefits. Adhered and non-adhered participants were completing their prescriptions, so dropouts were due to other reasons rather than challenges on understanding or following the prescriptions. Adherence to the walking intervention showed no relation to age or diagnose. Probably the type of exercise prescribed, walking, was not encouraging enough for some individuals and future interventions may include other types of physical activity (e.g., cycling, swimming, callisthenics), individually or as group-based activities (e.g., dancing, team sports, instruction-led group exercise), to improve aerobic and other physical capacities (e.g., strength, flexibility).

The CAMINEM Programme tools [22,26] were implemented and interdisciplinary work was achieved in the real setting, as previous experiences pointed out as a main challenge to cope with, such as lack of time, training and protocols [10,11,14,49]. Follow-up of the exercise prescriptions was possible using the CAMINEM Programme tools.

The CAMINEM procedures for implementation have been the framework to adapt in Catalonia the Swedish Physical Activity on Prescription model (FaR, in Swedish) under a 3^rd^ Health Programme grant by the European Commission [50].

### 4.5. Study Limitations

The CAMINEM practice-driven, rather than research-driven, procedures resulted in poor data collection. In the eighteen-month intervention period, the recruitment of patients was the responsibility of health practitioners and was undertaken during their regular consultations. Adoption of the programme relied on practitioners who were motivated to participate during their regular practice, even though they did not have formal recommendations from the healthcare provider. This resulted in a relatively low number of referrals and a high proportion of missing data related to clinical health outcomes, which limited the impact of the clinical results. Perhaps tighter control on data collection and referral procedures co-ordinated by the steering group may have shown more positive outcomes. Organisational instructions by decision-makers may have encouraged motivated practitioners and served as a guarantee for some practitioners who did not refer patients or only referred a few.

Both primary health-care settings are located in socially deprived neighbourhoods and some practitioners reported that health promotion interventions are rarely adopted by some ethnic minorities. The overall intervention was strongly associated with the personal interest of all agents involved, which had to cope with the political and financial situation which led to structural changes in the public administration (both Catalan and Spanish) in general and health-care administration in particular, such as, staff layoffs or budget reduction, among other organisational issues.

### 4.6. Practical Implications

This study assessed tools and procedures to encourage practitioners during their regular consultations to prescribe exercise to cope with the most prevalent chronic diseases.Regular practitioners had severe time constraints to strictly collect relevant health data to monitor the effects of the exercise prescriptions.The procedures to prescribe exercise were easy to understand and complete by the participants, easy to deliver, to monitor and to adapt by the exercise physiologists.The intervention framework may provide practice-based evidence to adapt exercise-on-prescriptions approaches in real settings to achieve public health goals.

## 5. Conclusions

The overall goal of this study was to evaluate the applicability of an exercise-on-prescription intervention in primary health-care settings: the CAMINEM Programme. CAMINEM used an interdisciplinary approach comprised of primary health-care practitioners (general practitioners and nurses) and exercise physiologists. The intervention was carried out with the underlying aim of improving regular patients’ health by facilitating the adoption and maintenance of exercise levels with safe, home-based exercises (walking). Two separate study objects provided the focus for this study. First, regular patients, with chronic health conditions that may benefit from exercise, that attended their primary health-care centre were assessed for intervention effects. On the other hand, both patients and health practitioners (i.e., general practitioners and nurses) were to be evaluated for intervention feasibility.

The CAMINEM Programme has been found to be feasible in real practice. Exercise prescriptions were for moderate-intensity aerobic activity in the form of walking outdoors following specific urban routes. It reached the target population, it was adopted by health practitioners, and it was implemented following the procedures. Health outcomes were positive although few clinical effects showed statistically significant differences. Health-related quality-of-life of adhered participants improved and they were less likely to visit their GP or nurse.

Future research-oriented initiatives may target selected health practitioners to collect all data or include an external researcher with this responsibility and use objective measures for physical activity levels, sedentary behaviour, and health-related fitness evaluation. All of the beforementioned requires specific funding and may not be feasible in real practice.

On the other hand, future practice-oriented interventions may design specific and safe procedures to prescribe, monitor, and follow-up home-based exercise to improve other fitness capacities (e.g., muscle strength, balance, flexibility) or to improve aerobic capacity by walking or other activities according to the participants’ preferences or accessibility. Selection of feasible fitness tests to apply in real practice may also be of interest.

## Figures and Tables

**Figure 1 ijerph-18-03192-f001:**
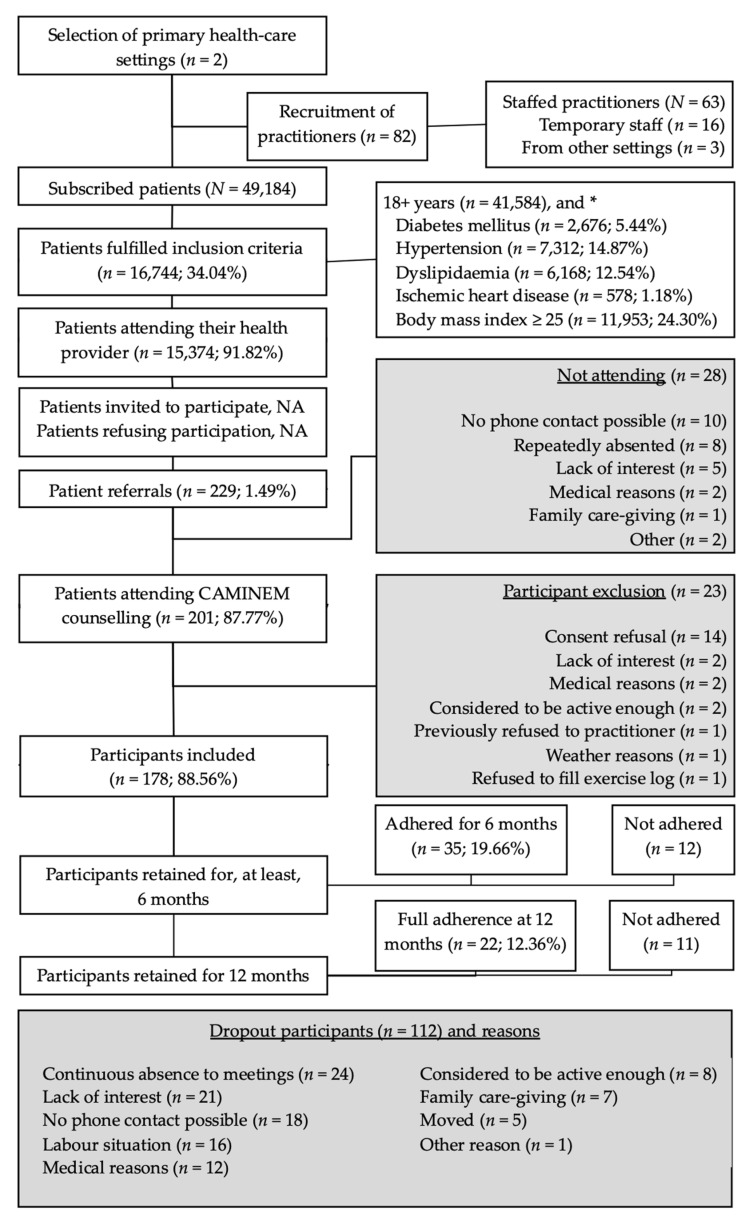
Flowchart of the participants included in the CAMINEM intervention. * Patients may have more than one condition. The CAMINEM Programme addressed patients from two primary health-care settings in socially disadvantaged neighbourhoods. All practitioners (physicians and community nurses) were invited to participate. Patients to be included had to be visited by their practitioner during a regular visit, accepted their oral invitation, being referred to the exercise physiologist and finally sign the informed consent. Retention refers to not drop out while adherence shows fulfillment of the exercise prescription.

**Figure 2 ijerph-18-03192-f002:**
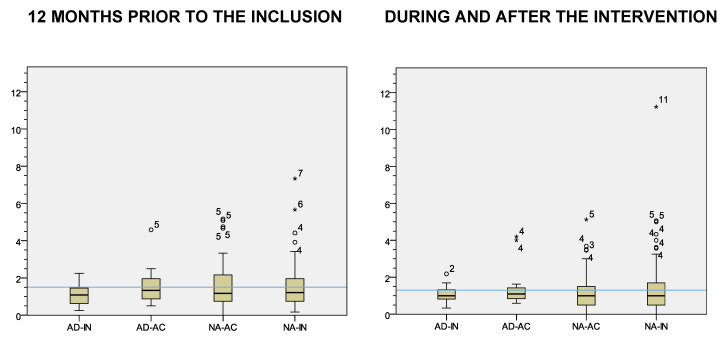
Health-care provider visits per month by intervention groups. *Note.* Blue lines = mean. AD-IN = adhered and previously inactive, AD-AC = adhered and previously active, NA-AC = non adhered and previously active, NA-IN = non adhered and previously inactive * stands for “Outliers”.

**Table 1 ijerph-18-03192-t001:** Reach, Effectiveness, Adoption, Implementation, and Maintenance (RE-AIM) Dimensions and their evaluative questions. Based on Glasgow [32] and Estabrooks & Gyurcsik [33].

RE-AIM Dimension	Evaluative Questions
Reach (Individual level)	What percentage of potentially eligible participants will take part and how representative are they?
Efficacy or Effectiveness (Individual level)	What impact did the intervention have on all participants who began the program, on process intermediate and primary outcomes, and on both positive and negative (unintended) outcomes including quality of life?
Adoption (Setting level)	What percentage of settings and intervention agents will participate and how representative are they?
Implementation (Both setting or agent and individual level)	To what extent are the various intervention components delivered as intended, especially when conducted by regular staff in applied settings?To what extent did the participants receive and enacts the intervention components?
Maintenance (Both setting and individual level)	To what extent are different intervention components continued or institutionalized? What are the long-term effects?

**Table 2 ijerph-18-03192-t002:** Participants grouping and reasons for prescription.

	GENDER				PRACTITIONER				SETTING				AGE GROUP						BASAL PA LEVEL ^a^				TOTAL	
	Male		Female		Physician		Nurse		A		B		18–44		45–64		65+		Insuffic.		Sufficient			
	*n* = 63	(%)	*n* = 115	(%)	*n* = 30	(%)	*n* = 148	(%)	*n* = 77	(%)	*n* = 101	(%)	*n* = 25	(%)	*n* = 92	(%)	*n* = 61	(%)	*n* = 91	(%)	*n* = 86	(%)	*N* = 178	(%)
Reasons for prescriptions ^b^																								
Overweight																								
	23	(36.5)	**67**	(58.3)	**17**	(56.7)	**73**	(49.9)	**37**	(48.1)	53	(52.5)	**21**	(84.0)	41	(44.6)	28	(45.9)	**54**	(59.3)	36	(41.9)	**90**	(50.6)
Hypertension																								
	**31**	(49.2)	49	(42.6)	9	(3.0)	71	(48.0)	26	(33.8)	**54**	(53.5)	3	(12.0)	**43**	(46.7)	**34**	(55.7)	39	(42.9)	**40**	(46.5)	80	(44.9)
Dyslipidemia																								
	27	(42.9)	37	(32.2)	10	(33.3)	54	(36.5)	22	(28.6)	42	(41.6)	8	(32.0)	35	(38.0)	21	(34.4)	33	(36.3)	31	(36.0)	64	(36.0)
Diabetes Mellitus																								
	29	(46.0)	28	(24.3)	3	(10.0)	54	(36.5)	17	(22.1)	40	(39.6)	2	(8.0)	33	(35.9)	22	(36.1)	30	(33.0)	27	(31.4)	57	(32.0)
Musculoskeletal																								
	5	(7.9)	34	(29.6)	10	(33.3)	29	(19.6)	14	(18.2)	25	(24.8)	3	(12.0)	19	(20.7)	17	(27.9)	22	(24.2)	17	(19.8)	39	(21.9)
Cardiovascular																								
	17	(27.0)	13	(11.3)	5	(16.7)	25	(16.9)	9	(11.7)	21	(20.8)	2	(8.0)	9	(9.8)	19	(31.1)	16	(17.6)	14	(16.3)	30	(16.9)
Respiratory																								
	15	(23.8)	9	(7.8)	5	(16.7)	19	(12.8)	6	(7.8)	18	(17.8)	1	(4.0)	17	(18.5)	6	(9.8)	11	(12.1)	13	(15.1)	24	(13.5)
Mental illness																								
	2	(3.2)	17	(14.8)	5	(16.7)	14	(9.5)	7	(9.1)	12	(11.9)	1	(4.0)	12	(13.0)	6	(9.8)	11	(12.1)	8	(9.3)	19	(10.7)
Other																								
	5	(7.9)	8	(7.0)	2	(6.7)	11	(7.4)	4	(5.2)	9	(8.9)	1	(4.0)	10	(10.9)	2	(3.3)	8	(8.8)	5	(5.8)	13	(7.3)
Number of health conditions																								
1	16	(25.4)	**37**	(32.2)	9	(30.0)	**44**	(29.7)	**35**	(45.4)	18	(17.8)	**11**	(44.0)	**27**	(29.3)	15	(24.6)	24	(26.3)	**28**	(32.5)	**53**	(29.8)
2	17	(27.0)	34	(29.6)	**11**	(36.7)	40	(27.0)	24	(31.2)	27	(26.7)	**11**	(44.0)	25	(27.2)	**51**	(28.7)	25	(27.5)	26	(30.2)	51	(28.7)
3	**18**	(28.6)	28	(24.3)	7	(23.3)	39	(26.4)	13	(16.9)	**33**	(32.7)	3	(12.0)	26	(28.3)	46	(25.8)	**26**	(28.6)	20	(23.3)	46	(25.8)
4+	12	(19.0)	16	(13.9)	3	(10.0)	25	(16.9)	5	(6.5)	23	(22.8)	0	(0.0)	14	(15.2)	28	(15.7)	16	(17.6)	12	(14.0)	28	(15.7)

Note. PA = physical activity; Insuffic. = Insufficient. Bold: the most common condition. ^a^ One missing. ^b^ The total sums exceed 100%.

**Table 3 ijerph-18-03192-t003:** Homogeneity for continuous variables at baseline among groups.

GROUP ^a^	AD-IN (*n* = 15)		AD-AC (*n* = 20)		NA-AC (*n* = 66)		NA-IN (*n* = 76)		*p*-value
	M	SD	M	SD	M	SD	M	SD	
Age *	56.0	9.2	64.5	6.5	59.1	13.1	55.8	12.6	0.029
Clinical parameters									
BMI (kg/m^2^) *	32.5	4.2	30.2	4.3	30.9	4.3	33.2	5.2	0.021
Waist circumference (cm)	111.4	15.3	102.3	8.0	105.7	11.4	106.0	9.8	0.487
Systolic blood pressure (mmHg)	131.1	11.2	140.3	15.3	134.3	17.0	134.2	15.7	0.385
Diastolic blood pressure (mmHg)	79.2	8.6	77.1	7.4	77.6	11.0	79.8	10.8	0.575
Resting heart rate (beats/min)	74.0	14.1	75.2	15.8	73.0	12.5	77.0	11.5	0.552
Total blood cholesterol (mg/dl)	201.9	26.8	203.8	40.5	202.7	41.5	207.5	38.9	0.923
LDL blood cholesterol (mg/dl)	125.5	23.1	121.5	34.4	120.2	35.8	122.2	32.6	0.974
Triglyceride (mg/dl)	144.2	69.8	155.4	88.5	155.5	94.3	153.9	70.6	0.979
HDL blood cholesterol (mg/dl)	60.4	19.7	51.3	13.2	52.4	14.2	53.5	13.0	0.423
Plasma glucose (mg/dl)	115.2	33.5	114.5	25.5	116.9	42.8	118.9	38.9	0.970
Glycated haemoglobin (%) ^b^	7.5	1.5	7.1	0.9	7.7	1.4	7.4	1.2	0.693
SF-12v2 questionnaire outcomes									
Physical functioning	44.7	8.0	54.5	4.5	45.6	7.5	45.1	9.0	0.184
Role physical	46.1	5.3	57.4	4.7	46.0	10.4	44.2	12.8	0.161
Bodily pain	33.4	5.9	25.3	4.6	33.7	11.6	36.4	12.9	0.272
General health	40.4	5.5	48.2	9.9	40.9	6.9	40.1	8.1	0.237
Vitality	46.0	11.3	54.2	21.0	52.9	11.6	51.3	13.8	0.547
Social functioning	41.2	9.8	51.8	10.6	46.8	12.4	44.6	12.9	0.448
Mental health	45.5	10.7	53.9	8.4	45.4	11.8	42.5	12.4	0.251
Role emotional	50.3	13.5	47.2	10.6	49.2	11.8	50.1	12.7	0.954
PCS: Summary scale Physical	39.3	5.5	45.1	2.2	39.5	6.1	39.7	6.2	0.353
MCS: Summary scale Mental	48.4	14.0	53.5	9.3	51.2	13.3	49.4	15.3	0.871
Health-care attendance									
In the year prior to inclusion (visits per month)	1.1	0.6	1.5	0.9	1.6	1.2	1.5	1.2	0.574

Note. AD-IN = adhered and previously inactive, AD-AC = adhered and previously active, NA-AC = non adhered and previously active, NA-IN = non adhered and previously inactive, BMI = body mass index. * *p* < 0.05. ^a^ 1 missing. ^b^ Sample size for participants diagnosed with diabetes mellitus: Total (*n* = 57), AD-IN (*n* = 6), AD-AC (*n* = 9), NA-AC (*n* = 18), NA-IN (*n* = 24).

**Table 4 ijerph-18-03192-t004:** Homogeneity at baseline (M1) between groups by adherence.

GROUPS	AD-Group (*n* = 35)		NA-Group (*n* = 142)		*p*-value
	M	SD	M	SD	
Age	60.8	8.8	57.4	12.9	0.126
Clinical parameters					
BMI (kg/m^2^)	31.2	4.3	32.2	4.9	0.466
GROUPS	AD-Group		NA-Group		*p*-value
	*n =* 35	(%)	*n =* 142	(%)	
Diagnosed dyslipidaemia	19	(29.7)	45	(70.3)	0.022 *^,a^
Diagnosed overweight	12	(13.3)	78	(86.7)	0.046 *^,b^

Note. AD-Group = adhered, NA-Group = non-adhered, BMI = body mass index, kg = kilogram, m = metre. ^a^ Continuity correction = 5.270. ^b^ Continuity correction = 3.998. * *p* < 0.05.

**Table 5 ijerph-18-03192-t005:** Intention-to-treat multivariate analyses of variance

Measure		M1		MOT3		Signification effects		Evolution	
Group		AD (*n* = 35)	NA (*n* = 142)	AD (*n* = 23)	NA (*n* = 110)	Group	Time	AD	NA
HR ^a^ (beats/min)	M	72.9	73.6	72.7	76.2			0.21%	−3.50%
	SD	15.8	12.1	15.7	13.4	*p* = 0.583	*p* = 0.564	*p* = 0.969	*p* = 0.271
	*n*	14	32	14	32	Mdif = 2.1	Mdif = 1.3	Mdif = 0.1	Mdif = 2.7
CHO ^a^ (mg/dl)	M	201.2	222.3	193.4	199.0			4.06%	11.71%
	SD	25.4	43.3	36.4	41.8	*p* = 0.287	*p* = 0.008**	*p* = 0.383	*p* = 0.001**
	*n*	13	23	13	23	Mdif = 13.3	Mdif = 15.6 95% CI [4.3, 26.9]	Mdif = 7.8	Mdif = 23.3 95% CI [9.7, 36.9]
LDL ^a^ (mg/dl)	M	125.8	127.7	127.7	115.1			−1.48%	10.95%
	SD	22.2	35.0	40.4	38.4	*p* = 0.698	*p* = 0.413	*p* = 0.852	*p* = 0.131
	*n*	9	14	9	14	Mdif = 5.4	Mdif = 5.4	Mdif = 1.9	Mdif = 12.6
BMI ^b^ (kg/m^2^)	M	30.5	33.4	29.9	32.8	M1	MOT3	1.99%	1.91%
	SD	4.1	4.6	4.0	4.4	*U* = 1553.0	*U* = 242.0	*Z* = 1.862	*Z* = 3.314
	*n*	18	44	18	44	*p* = 0.230	*p* = 0.013*	*p* = 0.063	*p* = 0.002**
WC ^b^ (cm)	M	102.6	108.7	100.6	106.75	M1	MOT3	1.99%	1.87%
	SD	9.09	12.3	7.2	11.3	*U* = 441.5	*U* = 75.0	*Z* = 0.847	*Z* = 1.791
	*n*	7	8	7	8	*p* = 0.995	*p* = 0.521	*p* = 0.397	*p* = 0.073
SBP ^b^ (mmHg)	M	137.5	135.3	138.6	130.8	M1	MOT3	−0.76%	3.50%
	SD	15.6	15.9	16.9	14.6	*U* = 1775.0	*U* = 375.0	*Z* = 0.207	*Z* = 2.349
	*n*	17	51	17	51	*p* = 0.303	*p* = 0.124	*p* = 0.836	*p* = 0.019*
DBP ^b^ (mmHg)	M	76.8	80.4	76.2	79.4	M1	MOT3	0.76%	1.31%
	SD	6.9	11.1	9.1	10.3	*U* = 1943.0	*U* = 385.5	*Z* = 0.649	*Z* = 0.791
	*n*	17	51	17	51	*p* = 0.757	*p* = 0.161	*p* = 0.516	*p* = 0.429
TG ^b^ (mg/dl)	M	144.9	171.2	112.5	145.9	M1	MOT3	28.77%	17.37%
	SD	70.9	90.4	43.4	87.8	*U* = 1482.5	*U* = 206.5	*Z* = 2.621	*Z* = 1.773
	*n*	13	21	13	21	*p* = 0.736	*p* = 0.220	*p* = 0.009**	*p* = 0.076
HDL ^b^ (mg/dl)	M	51.9	51.2	52.0	50.0	M1	MOT3	−0.21%	2.42%
	SD	13.0	12.1	8.5	11.1	*U* = 1113.5	*U* = 132.5	*Z* = 0.178	*Z* = 0.624
	*n*	9	14	9	14	*p* = 0.808	*p* = 0.864	*p* = 0.859	*p* = 0.533
GLY ^b^ (mg/dl)	M	114.4	119.1	107.2	113.0	M1	MOT3	6.67%	5.43%
	SD	28.9	41.7	23.5	25.3	*U* = 1701.0	*U* = 243.5	*Z* = 0.754	*Z* = 0.505
	*n*	14	21	14	21	*p* = 0.983	*p* = 0.448	*p* = 0.451	*p* = 0.614
HbA_1c_ ^b,c^ (%)	M	7.4	7.4	7.2	7.2	M1	MOT3	2.58%	1.77%
	SD	1.4	0.9	1.0	1.1	*U* = 214.5	*U* = 33.5	*Z* = 0.406	*Z* = 0.254
	*n*	7	7	7	7	*p* = 0.565	*p* = 0.565	*p* = 0.684	*p* = 0.799

Note. M1 = basal, MOT3 = over time post-intervention, AD = adhered to the intervention, NA = non adhered to the intervention, HR = heart rate, min = minute, Mdif = mean difference, CHO = total cholesterol, mg = milligram, dl = decilitre, CI = confidence interval, BMI = body mass index, kg = kilogram, m = metre, WC = waist circumference, cm = centimetre, SBP = systolic blood pressure, mmHg = millimetres of mercury, DBP = diastolic blood pressure, TG = triglyceride, GLY = fasting glucose. ^a^ 2 × 2 multivariate analysis. Age-adjusted comparison. ^b^ Mann-Whitney *U* (pre- and post-intervention) and Wilcoxon’s *Z* (pre- and post-intervention within group). ^c^ Percentage varies due to decimals. * *p* < 0.05 ** *p* < 0.01.

**Table 6 ijerph-18-03192-t006:** Distribution of CAMINEM adherence by intervention groups.

	ATTENDANCE		RETENTION		COMPLIANCE		
	Irregular	Regular	<180 d	≥ 180 d	ExRx not completed	ExRx completed	TOTAL ^a^
GROUP	*n* = 66 (%)	*n* = 111 (%)	*n* = 130 (%)	*n* = 47 (%)	*n* = 82 (%)	*n* = 95 (%)	*n* = 177 (%)
AD-IN	0 (0)	15 (100)	0 (0)	15 (100)	0 (0)	15 (100)	15 (100)
AD-AC	0 (0)	20 (100)	0 (0)	20 (100)	0 (0)	20 (100)	20 (100)
NA-AC	27 (40.9)	39 (59.1)	61 (92.4)	5 (7.6)	34 (51.5)	32 (48.5)	66 (100)
NA-IN	39 (51.3)	37 (48.7)	69 (90.8)	7 (9.2)	48 (63.2)	28 (36.8)	76 (100)

Note. ExRx = exercise prescription, AD-IN = adhered and previously inactive, AD-AC = adhered and previously active, NA-AC, not adhered and previously active, NA-IN = not adhered and previously inactive. ^a^ 1 missing.

## Data Availability

Restrictions apply to the availability of these data. Data was obtained from Institut Català de la Salut and are available on request from the corresponding authors with the permission of Institut Català de la Salut.

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
