# Peer review of "Feasibility Assessment of the Let’s Walk Programme (CAMINEM): Exercise Training and Health Promotion in Primary Health-Care Settings"

_ijerph, 2021, doi:10.3390/ijerph18063192_

Round 1
Reviewer 1 Report
The manuscript entitled “Feasibility assessment of the Let’s Walk programme (CAMINEM): exercise training and health promotion in primary health-care setting” deals with an assessment of a physical program in Catalonia, Spain, on a group of 229 persons who have a certain chronic disease. The evaluation had the purpose of analyzing 5 different stages of the program application (reach, effectiveness, adoption, implementation and maintenance) among the regular population. Authors claim that is the first time that this kind of program is pragmatically assessed.
The manuscript cannot be considered as suitable for publication if the following comments are not correctly addressed:
- The manuscript lacks of clearness on the methodology and the results. It’s not clear what is the age range of the study group, and how this could influence on the general conclusions (e.g. if the study group is manly elder, what kind of approach might be carried out to encourage a walking exercise?)
- Some characteristics of the input data (walking number of hours daily/weekly, features of the exercise: indoor, outdoor etc.) are not clearly displayed. This is important because it could give the respective solution to enhance the walking program.
- Results show the biochemical references of the study group, but it’s not clear how this data is influencing the assessment of the implementation of the CAMINEM program. This results should be used for testing whether the programs is effective or not, but I couldn’t find any discussion regarding this subject.
- Also, it’s not clear whether the results are useful to analyze other situations on other regions of Spain, for instance. The study group is small and it’s not representing the behavior of other parts of Spain. A deep discussion on this must be carried out.
- The message “Error!...” is often presented throughout the manuscript, making even more difficult to follow up the idea of research.
Author Response
Please, see the attachment.

Reviewer 2 Report
査読
This study investigated the impact of different levels of adherence to the CAMINEM program on subsequent health promotion. The research is conducted appropriately, but there are many carelessness in writing the paper.
Major comments
1) Figure 1 and line 222: Please modify Figure 1 to show the clear relationship between the subject number of four groups groups and that in Figure 1.
2) Concerning the five dimensions, Results and Discussion section describes on reach, effects, adoption, and implementation on 3.1~3.4 and 4.1~4.4, respectively. But there's no description on maintenance in Results and Discussion section.
Minor comments
Line 114: Please add the phrase of “(Ask, Advise, Agree, Assist, Arrange)” after 5A.
Table 1: Align the heights of the dimension on the left and the question on the right to show the correspondence between them, please.
Line 142: Effects should be Efficacy or effectiveness.
Figure 1: Some of the letters in the table overlap each other, especially “(n=“. Please correct it.
Line 243: Table 1 should be Table 4.
Line 442: Supplementary files are not shown in the URL.
Author Response
Please, see the attachment.

Round 2
Reviewer 1 Report
The authors have correctly addressed my previous comments.
Author Response
Thank you for your time and effort to review our manuscript.
Reviewer 2 Report
Corrections have been made based on the comment of the reviewer, but the correction about the number of subjects is insufficient.
There are descriptions of "Participants included (178)" in Figure 1 and "TOTAL n=178" in Table 2. But the sum of "Basal PA level Insuffic (91)+suffic(86)" is 177 in Table 2.
On the other hand the sum of "AD-IN(15)+AD-AC(20)+NA-AC(66)+NA-IN(76)" is 177 in Table 3.
Author Response
Please, see the attachment.
